# Oral ascorbic acid 2-glucoside prevents coordination disorder induced via laser-induced shock waves in rat brain

Takaaki Maekawa[1,2☯]*, Takahiro Uchida[1,3☯], Yuka Nakata-Horiuchi[1], Hiroaki Kobayashi[1], Satoko Kawauchi[4], Manabu Kinoshita[5], Daizoh Saitoh[6], Shunichi Sato[4]

**1** Military Medicine Research Unit, Test and Evaluation Command, Japan Ground Self-Defense Force, Setagaya, Tokyo, Japan, **2** Division of Hematology, Department of Internal Medicine, National Defense Medical College, Tokorozawa, Saitama, Japan, **3** Department of Nephrology, Tokyo Medical University Hachioji Medical Center, Hachioji, Tokyo, Japan, **4** Division of Bioinformation and Therapeutic Systems, National Defense Medical College Research Institute, Tokorozawa, Saitama, Japan, **5** Department of Immunology and Microbiology, National Defense Medical College, Tokorozawa, Saitama, Japan, **6** Division of Traumatology, National Defense Medical College Research Institute, Tokorozawa, Japan

☯ These authors contributed equally to this work.
* tmaekawa-ndmc@umin.ac.jp

**Data Availability Statement:** All relevant data are within the paper and its Supporting Information files.

## Abstract

Oxidative stress is considered to be involved in the pathogenesis of primary blast-related traumatic brain injury (bTBI). We evaluated the effects of ascorbic acid 2-glucoside (AA2G), a well-known antioxidant, to control oxidative stress in rat brain exposed to laser-induced shock waves (LISWs). The design consisted of a controlled animal study using male 10-week-old Sprague-Dawley rats. The study was conducted at the University research laboratory. Low-impulse (54 Pa•s) LISWs were transcranially applied to rat brain. Rats were randomized to control group (anesthesia and head shaving, n = 10), LISW group (anesthesia, head shaving and LISW application, n = 10) or LISW + post AA2G group (AA2G administration after LISW application, n = 10) in the first study. In another study, rats were randomized to control group (n = 10), LISW group (n = 10) or LISW + pre and post AA2G group (AA2G administration before and after LISW application, n = 10). The measured outcomes were as follows: (i) motor function assessed by accelerating rotarod test; (ii) levels of 8-hydroxy-2'-deoxyguanosine (8-OHdG), an oxidative stress marker; (iii) ascorbic acid in each group of rats. Ascorbic acid levels were significantly decreased and 8-OHdG levels were significantly increased in the cerebellum of the LISW group. Motor coordination disorder was also observed in the group. Prophylactic AA2G administration significantly increased the ascorbic acid levels, reduced oxidative stress and mitigated the motor dysfunction. In contrast, the effects of therapeutic AA2G administration alone were limited. The results suggest that the prophylactic administration of ascorbic acid can reduce shock wave-related oxidative stress and prevented motor dysfunction in rats.

**Funding:** The authors received no specific funding for this work.

**Competing interests:** The authors have declared that no competing interests exist.

## Introduction

Blast injury has been a constant threat in recent conflicts because of frequent terrorist attacks using weapons such as improvised explosive devices (IEDs). These changes of combat style affect both civilian and military populations and result in a large number of patients suffering from blast-related traumatic brain injury (bTBI) [1–3].

Notably, most bTBI patients who are exposed to low or mild blast exposure lack any external physical evidences and abnormalities detected by conventional imaging devices; therefore, they are being diagnosed as having blast-related mild TBI (bmTBI). However, during the chronic phase, they develop persistent physiological and psychological changes associated with higher-order brain dysfunction during the chronic phase [4, 5]. The precise mechanisms of such bmTBI are complicated; however, studies have shown that blood-brain barrier (BBB) disruption [6], brain edema [7], and neuroinflammation [8] play certain roles in the development of bmTBI. It was also reported that oxidative stress was involved in the pathogenesis of bmTBI [9, 10] and that an antioxidant ameliorated behavioral deficits in a bmTBI model [11]. Oxidative stress has been implicated in multiple models of TBI [12, 13] and is mainly induced by reactive oxidative species (ROS) such as superoxide, hydroxyl radical, and hydrogen peroxide [14, 15]. While basal levels of ROS are present during normal redox reactions and the electron transport chain, excess amounts, as seen after injury, can be harmful. NADPH oxidase (NOX) is a multi-subunit enzyme that catalyzes the formation of superoxide radicals from available molecular oxygen. NOX is upregulated in multiple brain regions following blast injury, and neurons maximally contributes to a higher increase in the hippocampus compared with other neural cells [16, 17]. Moreover, oxidative stress also contributes to enhanced BBB permeability during bTBI via a pathway that involves increased matrix metalloproteinase activation [18].

Hydrogen sulfide, which is one of the antioxidants, also affected the controlled cortical impact injury in rats, leading to improved neurologic dysfunction, increased activities of endogenous antioxidant enzymes (superoxide dismutase and catalase), decreased levels of oxidative products (malondialdehyde and 8-iso-prostaglandin F2$\alpha$), increased BBB permeability, and attenuation of brain edema. Furthermore, the use of the $K_{ATP}$ channel blocker, 5-hydroxydecanoate, further demonstrated activation of mitochondrial adenosine triphosphate sensitive potassium channels, and oxidative stress was reduced following treatment with exogenous hydrogen sulfide [19, 20].

To investigate blast injuries, we used laser-induced shock waves (LISWs), which have advantages including safety, ease to use, compact device, versatility, and highly controllable shock wave energy [21], as compared with other shock wave sources such as micro actual explosions [22] and shock [23, 24]/blast [25] tubes. It is believed that the above-described bmTBI symptoms are associated with the primary mechanism, that is, the effects of the blast shock wave. Since LISWs are pure shock waves with no dynamic pressure component, LISWs could represent a useful tool to investigate the primary mechanisms of blast injuries.

In this study, considering the ability to exert strong antioxidant activity by acting as free radical scavenger [26–29], we evaluated the effects of ascorbic acid 2-glucoside (AA2G), a stable derivative of ascorbic acid which is generated by binding glucose to conventional ascorbic acid [30], on the LISW-induced TBI model.

## Materials and methods

### Animals

Male 10-week-old Sprague-Dawley rats, weighing 310 to 380 g, were obtained from Japan SLC, Inc. (Shizuoka, Japan) and were provided with water and standard chow *ad libitum*.

They housed two per cage and maintained at a constant temperature (21o – 23˚C) and humidity (40–60%) with lights on 06:00–18:00. They were housed for 1 week in their home cages with a 12-h light-dark cycle before beginning any procedures. The rats were anesthetized with isoflurane (induction: 5%, maintenance: 2%) and their head were shaved to apply LISWs through intact scalp before the LISW application. All animals were closely monitored post-LISW application with weight and health surveillance recording as per IACUC guidelines. After the assessment of motor function, rats were perfused with saline and were sacrificed with decapitation by an experienced animal technician under deep anesthesia by intraperitoneal injection of 100 mg/kg pentobarbital sodium, and their brains were removed and were cut into cerebrum, hippocampus and cerebellum tissues. All animal experiments were conducted in accordance with the National Defense Medical College guidelines for the care and use of laboratory animals in research. The study protocol was approved by the Animal Ethics Committee of the National Defense Medical College (#16036).

## Administration of ascorbic acid

AA2G (Hayashibara Co., Okayama, Japan) was dissolved in distilled drinking water and was administered daily *per os* (p.o.) to rats at a dose of 250 mg/kg/day (250 mg of AA2G contained 125 mg/kg/day of ascorbic acid) after LISW application for a total of 7 days (LISW + post AA2G group). LISW group rats in this experiment also received p.o. pure distilled water after LISW application for 7 days (Fig 1A). Control group also received p.o. pure distilled water for the same period.

In another experiment, AA2G was administered daily to rats from 3 days before LISW application to 7 days after the LISW for a total of 10 days (LISW + pre and post AA2G group). Control and LISW groups received p.o. pure distilled water for the same period (Fig 1B).

## Generation and characteristics of LISW

The method for generation of an LISW is shown in Fig 2A. A laser target, which is a laser-absorbing material (0.5-mm-thick natural black rubber disk) on which an optically transparent material [1.0-mm-thick polyethyleneterephthalate (PET) sheet] is adhered, was placed on the tissue. The target was irradiated with a short laser pulse, which is absorbed by the rubber to induce a plasma, and its expansion is accompanied by a shock wave (LISW). This was a type of microexplosion process from a physics point of view. In the present study, the second harmonics of a Q-switched Nd:YAG laser (Brilliant b, Quantel, Les Ulis Cedex, France; wavelength, 532 nm; pulse width, 6 ns) was used, and the scheme was the same as those used for our previous studies [31–36]. Fig 2B shows typical temporal pressure profiles of LISWs generated at different laser fluences on the target. No pressure signals were detected beyond the time range (after 2 μs; Fig 2B). Shock wave energy of LISWs is highly controllable; the peak pressure increases monotonically with increasing laser fluence (Fig 2C) and the size of the wave source can be changed by changing the laser spot size on the target.

No dynamic pressure is produced in the generation of LISWs; therefore, the effects of acceleration or displacement can be excluded in animal studies, enabling analysis of the primary mechanism of bTBI. Positive pressure duration (hereafter simply termed duration) is an important parameter to examine shock wave to brain interactions. The duration of a typical IED explosion-related shock wave ranges from 200 μs to several milliseconds [37, 38]; however, the interaction of the human brain with such IED-related shock waves, especially brain to skull boundary effects, cannot be reproduced in the brains of small animals using when similar duration shock waves. Therefore, a scaling law that considers the anatomical differences between human brain and animal brains should be used. However, few studies have focused

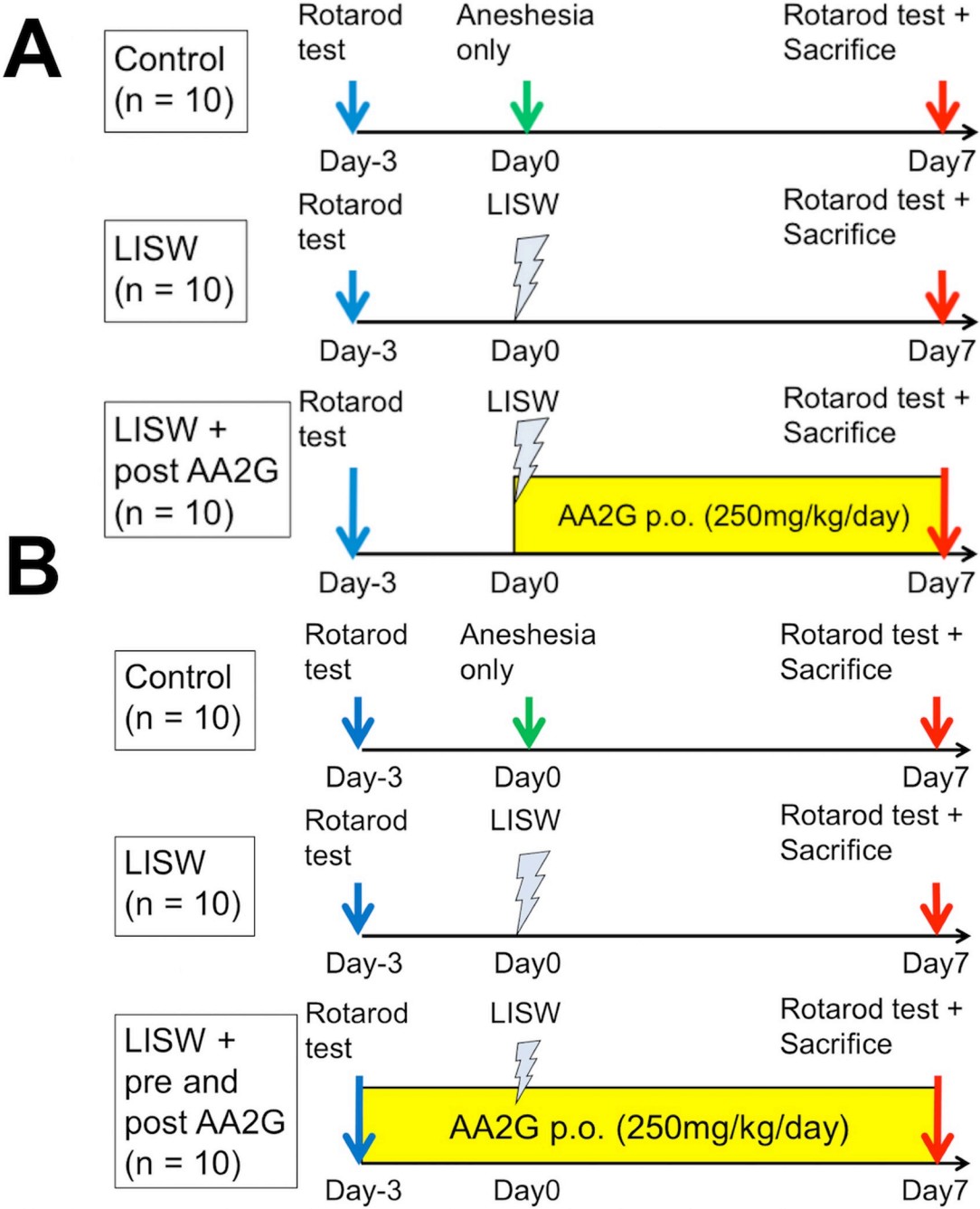

**Fig 1. The experimental protocol illustrating application of LISW and administration of AA2G.** (A) The experimental design showing therapeutic administration of ascorbic acid 2-glucoside (post AA2G). Rats were exposed to laser-induced shock wave (LISW) or anesthesia, and their motor function was assessed using a rotarod apparatus 3 day before and 7 days after the procedures. AA2G (250 mg/kg/day) or distilled water was administered only after the procedures for a total of 7 days. Male 10-week-old Sprague-Dawley rats were used (n = 10 in each experimental group). (B) The experimental protocol illustrating prophylactic and therapeutic administration of AA2G (pre and post AA2G). In this experiment, motor functions were assessed 3 days before and 7 days after the procedures. AA2G or distilled water was administered from 3 days before the procedures for a total of 10 days. Male 10-week-old Sprague-Dawley rats were used (n = 10 in each experimental group).

on this important issue. In the present study, we assumed that the impulse (time-integrated positive pressure component) of the shock wave was the primary parameter for determining brain injury. Alley previously estimated impulses of various types of IEDs as a function of

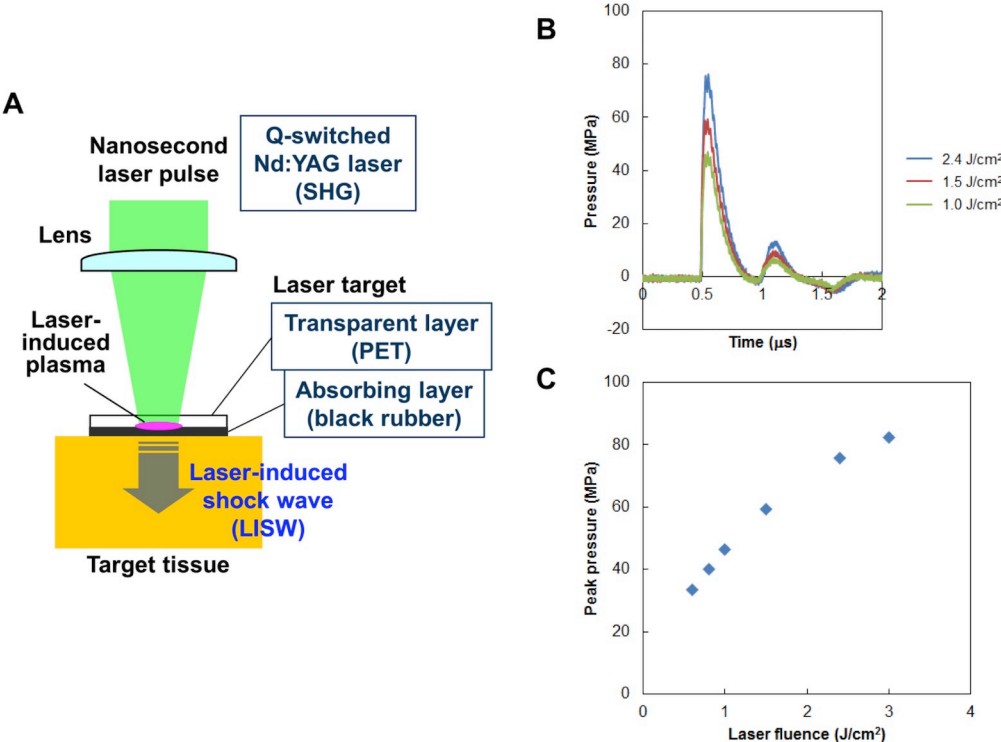

**Fig 2. Generation and characteristics of laser-induced shock waves (LISWs). (A)** Setup for generating a LISW. **(B)** Typical temporal waveforms of LISWs at different laser fluences on the laser target. **(C)** Dependence of peak pressure of LISW on laser fluence.

propagation distance ranging from ~6.9 to ~100 Pa•s for propagation distances of 1–10 m [37]. Although the duration of LISWs is approximately two to three orders of magnitude shorter than that of typical IED shock waves, their impulses can be easily controlled by changing the laser fluence and hence the peak pressure of LISW. IED impulses within this range can be replicated with LISWs. The limitations associated with this assumption are described in the Discussion and Limitations section.

## LISW application

The rats were anesthetized with isoflurane (induction: 5%, maintenance: 2%) and their head were shaved to apply LISWs through intact scalp. A laser target (a rubber disk of 8 mm in diameter covered with a PET sheet) was held with forceps and placed on the head skin of the rats; ultrasound gel (Hitachi Aloka Medical, Tokyo, Japan) was used between the bottom of the rubber and the head skin for acoustic impedance matching. For each application, rats were subjected to a set of four LISW pulses on the following sites of the scalp to cover the whole brain: 5 mm bilateral from a point at the intersection of auricular line with mid-sagittal line and then 5 mm anterior from each point. The laser spot size and fluence on the target were kept constant at 3 mm and 2.4 J/cm$^2$, respectively, producing an LISW with an impulse of 54 Pa•s. Mishra et al. examined physiological and pathological changes, as well as changes in blood-borne biomarkers in rats exposed to blasts with a wide range of overpressure and impulses using a shock tube [17]. They determined the conditions under which mild, moderate, severe, and lethal bTBIs were induced, and postulated that mild bTBI was caused by blasts

with overpressures ≤145 kPa, with a corresponding impulse of 250 Pa•s. The impulse used in this study (54 Pa•s) was within this range. The control rats were subjected to the same procedures without receiving LISW (sham injury).

## Assessment of motor function

Before and seven days after LISW application or sham injury, rats were tested using a rotarod apparatus (Penlab, Barcelona, Spain), by which abnormalities in pathways responsible for integrated vestibulomotor and sensorimotor function can be evaluated [39]. Motor function of the rats was assessed by measuring the latency (in second) during which the rats remained on a 60-mm-diameter rod rotating with an initial velocity of 4 rpm and an acceleration of 8 rpm. After one acclimatization session on the rotarod apparatus, rats received five trials at 5-minute intervals. Each trial was terminated when the rat fell completely off the rod or gripped the rod and rotated in a (spun around one) complete revolution and time on the rod was recorded. The trials with shortest and longest time on the rod were eliminated and the average time on the rod of the three remaining trials was used for statistical analyses, according to previously published studies [40, 41].

## Protein extraction

After the assessment of motor function, rats were perfused with saline and were sacrificed by decapitation under deep anesthesia by intraperitoneal injection of 100 mg/kg pentobarbital sodium, and their brains were removed and were cut into cerebrum, hippocampus and cerebellum tissues. At this point, it was observed that none of rats used showed apparent gross hemorrhage. Protein samples of the obtained brain tissues were extracted in a cell lysis buffer (RIPA Buffer, Wako, Osaka, Japan) containing 1% Protease Inhibitor Cocktail (NACALAI TESQUE, Kyoto, Japan). Concentrations of extracted protein samples were determined using BCA protein assay kits (Pierce, Rockford, IL, USA).

## Assessment of oxidative stress and cytokine expression

Levels of 8-hydroxy-2'-deoxyguanosine (8-OHdG), an oxidative stress marker, and tumor necrosis factor-α (TNF-α) were measured using ELISA kits (8-OHdG: Japan Institute for the Control of Aging, NIKKEN SEIL Co., Shizuoka, Japan; TNF-α: R & D Systems, Minneapolis, MN, USA).

## Measurements of ascorbic acid levels in cerebellum and plasma

We focused on cerebellum because 8-OHdG levels in cerebellum were noticeably increased by LISW exposure. Cerebellum tissues (60 mg) were homogenized in 5% metaphosphoric acid (840 mg) and centrifuged at 10000 g for 15 min at 4˚C, essentially as previously described [42, 43]. Plasma samples were mixed with equal amount of 10% metaphosphoric acid and centrifuged at 21000 g for 10 min at 4˚C. Thereafter, ascorbic acid levels of the obtained samples were measured by Hayashibara Co., using high performance liquid chromatography, prominence UFLC LC-20AD (Shimadzu Co, Kyoto, Japan).

## Dihydroethidium (DHE) staining

To evaluate reactive oxygen species (ROS) formation in the tissues, cryostat sections of fresh-frozen cerebellum tissues were subjected to DHE (Invitrogen, Waltham, MA, USA) staining, essentially as previously described [11, 44]. Brains were cryosectioned (4 μm) onto glass slides, rinsed in pure distilled water for 5 minutes, and incubated in 5μM of DHE for 30 minutes in

the dark. The oxidation product, ethidium, is formed from DHE by ROS resulting in ethidium accumulation within cells producing ROS. Slides were subsequently rinsed by PBS and ethidium was detected as red nuclei. Images of each section were obtained with a digital camera at a magnification of 200×. Thereafter, mean fluorescence intensity (MFI) was evaluated by measuring and averaging the immunofluorescence intensities of 10 randomly selected cells under the same conditions using the image analysis software (LuminaVision ver. 2.04, Mitani Corporation, Tokyo, Japan).

## Statistics

Data are expressed as the mean ± SEM. Differences between two experimental groups were assessed using Mann–Whitney test, and those among more than three experimental groups were assessed by one-way ANOVA with the Tukey HSD post hoc test. All statistical analyses were conducted with a significance level of $\alpha = 0.05$ ($P < 0.05$) using JMP software (version 11; SAS Institute Inc., Cary, NC).

## Results

### LISW-induced TBI involves oxidative stress in the brains

First, we measured levels of 8-OHdG and TNF-α to investigate the involvement of oxidative stress and inflammatory cytokine, respectively, in the brain of this LISW-induced TBI model. As shown in Fig 3A, 8-OHdG levels in cerebrum, hippocampus, and cerebellum tissues of the LISW group were all significantly higher than those of the control group. Among them, cerebellum 8-OHdG levels in the control group were undetectable, and the difference of 8-OHdG levels between the two groups was the most prominent. On the other hand, TNF-α levels did not differ between the two groups in any of the brain regions at least in the limited cases (Fig 3B). These results showed that this LISW-induced TBI model was characterized by oxidative stress in the brains and prompted us to evaluate the effects of AA2G on the model, especially focusing on cerebellar oxidative stress.

### Post treatment with ascorbic acid does not exert significant therapeutic effects on LISW-induced TBI

The time during which the rats remained on the rod of the rotarod, which was assessed before LISW application, did not differ among the control group, the LISW group and the LISW + post AA2G group (Fig 4A). The time in the LISW group, which was assessed seven days after LISW application, was significantly shorter than that in the control group. In contrast, the time in the LISW + post AA2G group, which was assessed seven days after LISW application, tended to be longer than the LISW group; however, the difference between the two groups did not reach statistical significance (Fig 4A).

We also evaluated cerebellar and plasma 8-OHdG levels in the LISW and the LISW + post AA2G groups. There was no statistical significance in cerebellar 8-OHdG levels between the LISW group and the LISW + post AA2G group (Fig 4B). 8-OHdG levels in plasma also did not differ between the two groups (Fig 4B). Ascorbic acid, if administered after LISW application, showed not significant therapeutic effects on LISW-induced bmTBI.

### Preventive ascorbic acid administration reduces cerebellar oxidative stress and improves motor dysfunction

As shown in Fig 5A, cerebellar ascorbic acid levels in the LISW group were significantly lower than those in the control group, whereas prophylactic administration of AA2G significantly

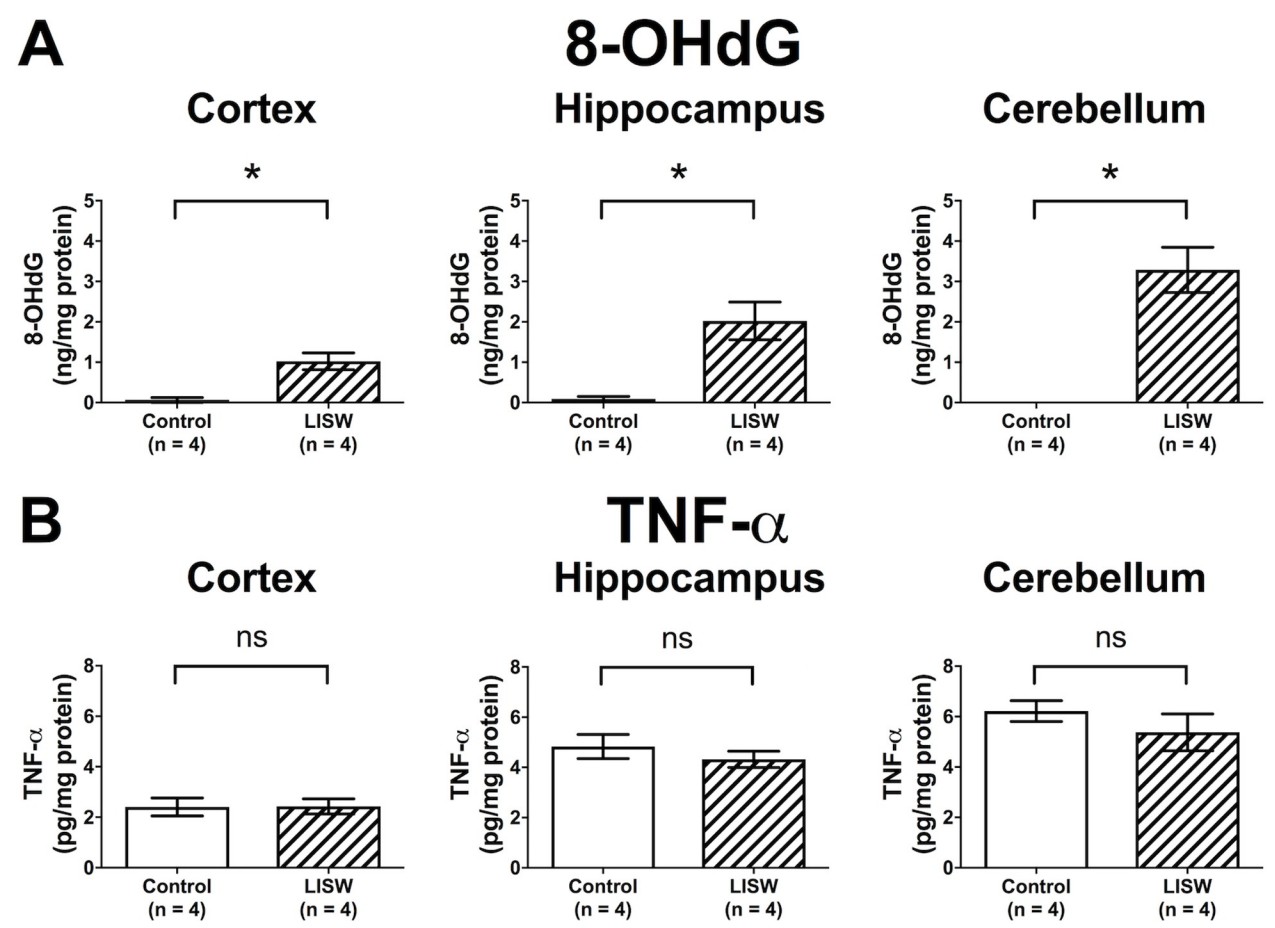

**Fig 3.** Levels of the following in the brain: (**A**) 8-hydroxy-2'-deoxyguanosine (8-OHdG) and (**B**) tumor necrosis factor-α (TNF-α) of the control and the LISW groups. Data are presented as the mean ± SEM (n = 4 in each group). Statistical significance was calculated by Mann–Whitney test. *P < 0.05, ns: not significant.

increased the levels. Interestingly, plasma levels of ascorbic acid in the LISW group were significantly higher than those in the control group, and prophylactic administration of ascorbic acid further increased the levels (Fig 5A).

Both cerebellar and plasma 8-OHdG levels in the LISW group were significantly higher than those in the control group; however, these levels were significantly decreased in the LISW + pre and post AA2G group (Fig 5B). These findings were supported by the results of DHE staining; ROS formation in cerebellum of the LISW group was significantly increased as compared to that of the control group, whereas the levels of ROS formation were significantly reduced in the LISW + pre and post AA2G group (Fig 5C).

Fig 5D shows the time during which the rats of each group remained on the rod of the rotarod. Although there was no statistical difference in the time among the three groups before LISW application (Fig 5D), the time in the LISW group, which was assessed seven days after LISW application, was significantly shorter than that in the control group (Fig 5D). On the other hand, the time in the LISW + pre and post AA2G group was significantly longer as compared to that in the LISW group. These results showed that prophylactic administration of AA2G increased cerebellar ascorbic acid levels, reduced ROS formation and resultant oxidative stress in cerebellum, and mitigated motor dysfunction of LISW-induced bmTBI in rats.

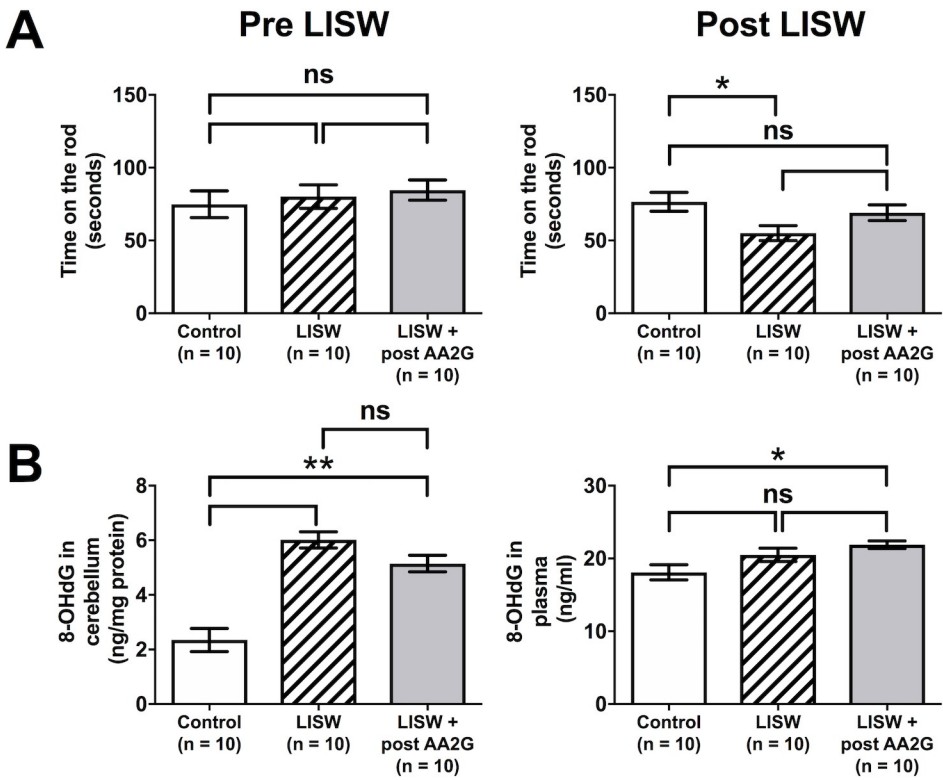

**Fig 4. The time remained on the rod and 8-OHdG level in cerebellum and plasma in the LISW group and the LISW + post AA2G group. (A)** The time remained on the rod during which the rats of the laser-induced shock wave (LISW) group and LISW + post ascorbic acid 2-glucoside (post AA2G) group, evaluated before LISW application (left) and after LISW exposure (right). **(B)** The 8-hydroxy-2'-deoxyguanosine (8-OHdG) levels in cerebellum (left) and plasma (right) in each group. Data are presented as the mean ± SEM (n = 10 in each group). Statistical significance was calculated by one-way ANOVA with the Tukey HSD post hoc test. $^*p < 0.05$, $^{**}p < 0.01$, ns: not significant.

## Discussion

The present study demonstrated that prophylactic AA2G administration ameliorated the motor coordination disorder due to the LISW-induced TBI (Fig 5D) through attenuation of the oxidative stress in the cortex and cerebellum (Fig 3A). Studlack et al. previously used beam walk, accelerating rotarod, rearing, open field, elevated plus maze, and light–dark box task tests in bTBI injury model rats to evaluate their motor function and anxiety-related behavior. They observed that the accelerating rotarod test exhibited significant differences between the bTBI and control groups from day 1 to day 14 post injury [45]. We considered that the prolonged symptoms were suitable for evaluating the efficacy of AA2G and confirmed that the accelerating rotarod test exhibited significant differences between the LISW and control groups in the present study. Expectedly AA2G restored the loss of cerebellar ascorbic acid levels, which was observed in the LISW group (Fig 5A). We examined the levels of TNF-α, one of the representative inflammatory cytokines in neuronal inflammation derived from monocytes and macrophages [8], in the brain. Although involvement of inflammatory pathway in the pathogenesis of bmTBI has been suggested [8], TNF-α levels in the brain were not elevated by LISW application in the present study (Fig 3B). The timing of TNF-α measurement might have affected the results.

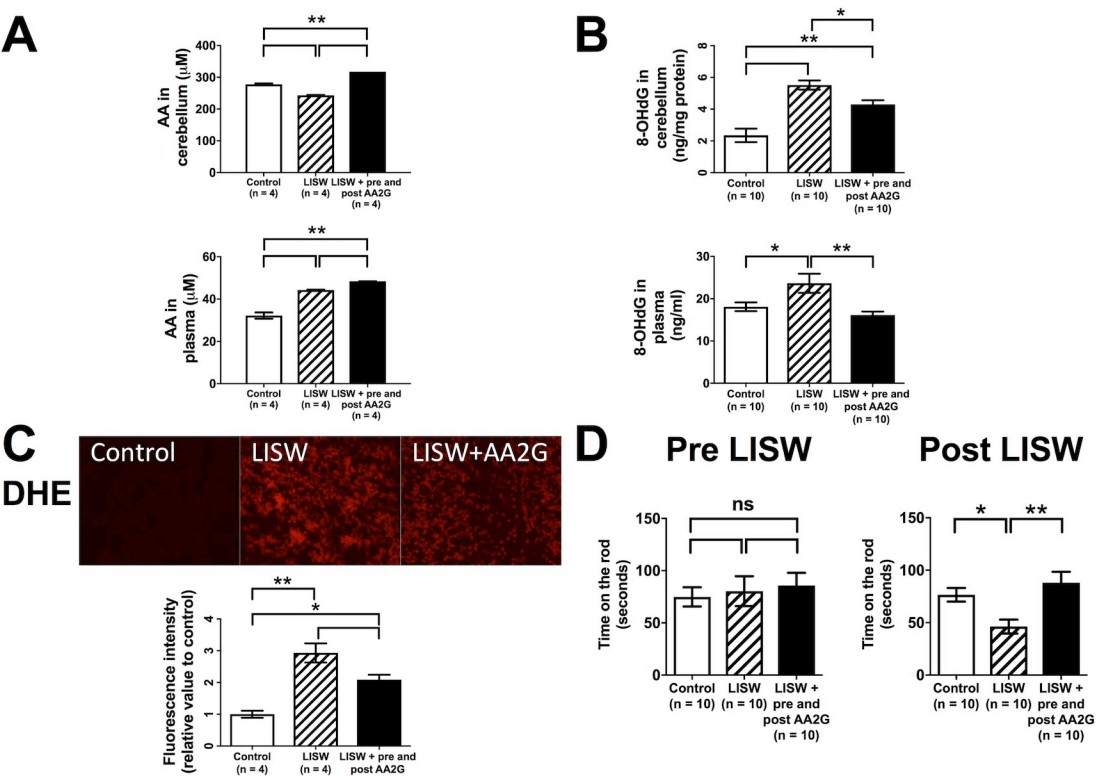

**Fig 5. AA levels in cerebellum and plasma, 8-OHdG levels in cerebellum and plasma, DHE staining of cerebellum, and the time remaining on the rod in the control group, the LISW group, and the LISW + pre and post AA2G group. (A)** Ascorbic acid (AA) levels in cerebellum (upper) and plasma (lower) of the control, laser-induced shock wave (LISW), and LISW + pre and post ascorbic acid 2-glucoside (AA2G) groups. **(B)** 8-hydroxy-2'-deoxyguanosine (8-OHdG) in cerebellum (upper) and plasma (lower) of each group. **(C)** Representative photomicrographs of dihydroethidium (DHE) staining of cerebellum from each group (upper). Relative fluorescence intensity of each group (lower). **(D)** The time during which the rats of each group remained on the rod, evaluated before LISW application (left) and after LISW exposure (right). Data are presented as the mean ± SEM (A and C; n = 4 in each group, B and D; n = 10 in each group). Statistical significance was calculated by one-way ANOVA with the Tukey HSD post hoc test. $^{*}$p < 0.05, $^{**}$p < 0.01, ns: not significant.

Ascorbic acid levels in cerebellum of the LISW group were significantly lower as compared to those of the control group, whereas the levels in plasma of the LISW group were significantly higher than those of the control group (Fig 5A). There seemed to be a discrepancy between local and systemic ascorbic acid levels. In this regard, we considered a possibility that ascorbic acid biosynthesis was increased in the LISW group to compensate the loss of brain ascorbic acid. However, it should be noted that such compensation cannot be applicable to humans, because rats can synthesize ascorbic acid *in vivo* but the capacity is lost in humans due to mutations in the gene encoding the last enzyme in ascorbic acid biosynthesis pathway [46].

Previous reports have demonstrated that the mechanisms of bTBI are unique and complex [24, 47–50]. Subsequent to brain damage that occurs at the immediate moment of exposure, the secondary effects yield a spectrum of injuries. It includes ROS production causing cellular damage through oxidation of cellular molecules such as DNA, proteins and lipids, which contributes to alterations in protein conformation and binding [11]. 8-OHdG is a stable product of oxidatively damaged DNA formed by hydroxyl radical, singlet oxygen and direct photodynamic action, and can be detected in tissue, serum, urine and other biomaterials [51]. Indeed,

in the present study, the levels of 8-OHdG were significantly elevated by LISW application. This ROS production may occur not only as a direct consequence of bmTBI but also as a result of secondary injury. For example, it was reported that the hemoglobin released from the disrupted red blood cells after blast exposure catalyzed free radical formation [52]. Further, the levels of oxidative stress and endogenous antioxidant proteins, such as superoxide dismutase, increased within several hours after blast exposure, and returned to normal levels within 5 days [24]. In addition, the elevation in the levels was directly proportional to the severity of injury [24]. Our previous study showed that single application of an LISW to the brain caused spreading depolarization, vasoconstriction and prolonged severe hypoxia in the cortex, which can induce neuronal cell alterations [21].

Ascorbic acid is a potent water soluble antioxidant in biological fluid [26] and has strong advantages for clinical application due to easy availability, low-cost, and minimal adverse effects. Indeed, various reports have shown its usefulness as a powerful antioxidant. For example, preventive and therapeutic potential of ascorbic acid in neurodegenerative diseases or chronic inflammatory diseases was reported [53, 54]. Administration of ascorbic acid also rescued irradiated mice from radiation-induced lethal gastrointestinal damage and reduced their mortality [41, 42, 55]. Importantly, no serious adverse effects related to ascorbic acid administration were reported in these studies. Further, AA2G is an ascorbic acid derivative that is stable in aqueous solution and barely induces cellular toxicity in cultured stem-cells, unlike ascorbic acid [56]. In addition, AA2G has protective effects against *Helicobacter pylori* infection in gastric epithelial cells [30] and confers protective effects on human sperm motility preservation through the freeze–thaw cycle [57]. In the present study, we evaluated the efficacy of AA2G in experimental LISW-induced TBI in rats at a dose of 250 mg/kg/day (this amount of AA2G contains 125 mg/kg/day of ascorbic acid). The dose of ascorbic acid was less than those had been used in the previous studies, demonstrating that ascorbic acid administration rescued irradiated mice [41, 42, 55]. However, because AA2G is hard to be resolved and considered to be more stable than conventional ascorbic acid [30], we chose our dosing at this level. Ascorbic acid effectively scavenges free radicals in vitro and might therefore be used as a radio-protectant that effectively scavenges ROS formed during radiation exposure [58]. There have been many reports regarding the protective effects of antioxidants given prior to exposure; however, there are few studies of successful post-exposure treatment [43, 59]. In present study, AA2G did not exert significant therapeutic effects on LISW-induced TBI when it was administered after LISW application; motor function assessed by rotarod apparatus and levels of oxidative stress did not significantly differ between the LISW and the LISW + post AA2G groups (Fig 4A). In line with this finding, in our previous studies [42, 43], treatment with ascorbic acid after radiation exposure could not also rescue lethal gastrointestinal damage in mice. In these studies, pre-treatment or post-treatment alone did not improve the survival of the mouse model; however, pre- and post-treatment significantly improved the survival. It is possible that scavenging ROS generated immediately after radiation by boosting the pretreatment with ascorbic acid may be necessary to improve the survival of irradiated mice. However, additional post-treatment with ascorbic acid also may be indispensable to further improve survival due to late or ongoing damage by oxidative stress.

We focused on cerebellar oxidative stress because 8-OHdG levels in cerebellum were noticeably increased by LISW exposure. Although the reason of the largest increase in cerebellar oxidative stress remains to be solved, motor coordination disorder, which can be associated with damage to the motor area of cortex and cerebellar, was actually observed in the LISW group. However, motor dysfunction is just a part of various symptoms which are induced by LISW-induced TBI. The involvement of oxidative stress and the efficacy of ascorbic acid in other symptoms of LISW-induced TBI should be carefully investigated.

In the present study, LISWs were applied only to the brain and were not accompanied by dynamic pressure. Therefore, it can be concluded that these results show the effects of the primary mechanism solely on the brain. Use of a shock tube would not have enabled us to completely exclude the effects of dynamic pressure or exposure to other part of the body; therefore, our results are useful to understand the mechanisms of primary bTBI. However, careful attention should be paid to the differences in characteristics between LISWs and actual blasts. Therefore, a comparative experimental study using a shock tube could give valuable insight into the mechanism and possible intervention for bTBI.

### Limitations

In the present study, there was only experimental data obtained at 7 days after LISW exposure, and longer follow-up was not done. The present study only used male and 10-week-old rats.

This study was performed based on the assumption that the impulse was the primary parameter to determine LISW-induced TBI. We recently identified methods to control the duration of LISW and are currently conducting experiments to examine the validity of this assumption. Therefore, the results obtained in the present study will be reevaluated after the validation.

### Conclusions

The present study showed that LISW-induced TBI involved ROS formation in the cerebellum and that prophylactic AA2G administration attenuated oxidative stress. In addition, prophylactic AA2G administration significantly ameliorated the motor coordination disorder induced by LISW-induced TBI. In contrast, therapeutic AA2G administration showed no significant effects. Our results indicated the oral AA2G could be a promising prophylactic agent for shock wave-related TBI.

### Supporting information

**S1 Data.**
(XLSX)

**S2 Data.**
(XLSX)

### Acknowledgments

We thank all staff members of the Military Medicine Research Unit for their help in conducting the present study. We are also very grateful for research team of Hayashibara Co., Japan, for measuring ascorbic acid concentration of the samples and kindly providing AA2G in all experiments.

### Author Contributions

**Conceptualization:** Takaaki Maekawa, Takahiro Uchida, Manabu Kinoshita, Shunichi Sato.

**Data curation:** Takaaki Maekawa.

**Formal analysis:** Takaaki Maekawa.

**Funding acquisition:** Shunichi Sato.

**Investigation:** Takahiro Uchida, Yuka Nakata-Horiuchi, Shunichi Sato.

**Methodology:** Shunichi Sato.

**Project administration:** Takaaki Maekawa, Takahiro Uchida, Yuka Nakata-Horiuchi, Hiroaki Kobayashi, Satoko Kawauchi, Manabu Kinoshita, Shunichi Sato.

**Resources:** Takahiro Uchida, Yuka Nakata-Horiuchi, Satoko Kawauchi, Manabu Kinoshita, Daizoh Saitoh, Shunichi Sato.

**Supervision:** Takahiro Uchida, Hiroaki Kobayashi, Satoko Kawauchi, Manabu Kinoshita, Daizoh Saitoh, Shunichi Sato.

**Writing – original draft:** Takaaki Maekawa, Takahiro Uchida.

**Writing – review & editing:** Takaaki Maekawa, Takahiro Uchida.

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
