## [Decision Letter · Decision Letter 0]

21 Nov 2019

PONE-D-19-25644

Oral ascorbic acid 2-glucoside prevents coordination disorder induced by blast-related mild traumatic brain injury in a rat model

PLOS ONE

Dear Dr Maekawa,

Thank you for submitting your manuscript to PLOS ONE. After careful consideration, we feel that it has merit but does not fully meet PLOS ONE’s publication criteria as it currently stands. Therefore, we invite you to submit a revised version of the manuscript that addresses the points raised during the review process.

Both reviewers feel that your study is sound and that your data support your conclusions, but the second review made some important recommendations that I agree would strengthen this submission significantly,

We would appreciate receiving your revised manuscript by Jan 05 2020 11:59PM. To enhance the reproducibility of your results, we recommend that if applicable you deposit your laboratory protocols in protocols.io, where a protocol can be assigned its own identifier (DOI) such that it can be cited independently in the future. For instructions see: http://journals.plos.org/plosone/s/submission-guidelines#loc-laboratory-protocols

We look forward to receiving your revised manuscript.

Kind regards,

Alfred S Lewin, Ph.D.

Academic Editor

PLOS ONE

Journal Requirements:

Additional Editor Comments (if provided):

Reviewers' comments:

Reviewer's Responses to Questions

**Comments to the Author**

1. Is the manuscript technically sound, and do the data support the conclusions?

Reviewer #1: Yes

Reviewer #2: Yes

2. Has the statistical analysis been performed appropriately and rigorously? 

Reviewer #1: Yes

Reviewer #2: Yes

3. Have the authors made all data underlying the findings in their manuscript fully available?

Reviewer #1: Yes

Reviewer #2: Yes

4. Is the manuscript presented in an intelligible fashion and written in standard English?

Reviewer #1: Yes

Reviewer #2: Yes

5. Review Comments to the Author

Reviewer #1: 'Oral ascorbic acid 2-glucoside prevents coordination disorder induced by blast-related mild traumatic brain injury in a rat model'. This is a well structred and well written manuscript. It is acceptable in its current form.

Reviewer #2: Comments

Introduction:

“Lines 62-65: Notably, most of the bTBI patients lack any external physical evidences and

abnormalities detected by conventional imaging devices, but in the chronic phase, they

develop persistent physiological and psychological changes associated with

higher-order brain dysfunction, which is called blast-related mild TBI (bmTBI) (4, 5).”

In reference to this statement, authors should specify that most of the bTBI patients who are exposed to low or mild blast exposure may not show signs of overt pathology in the acute phase. Mild blast should not be confused with mild TBI.

Lines 66-70: The cited work primarily evaluated the effects of shock wave (blast overpressure and/or underpressure) on neuropathology, oxidative stress, and the applicability of antioxidants in ameliorating the behavioral deficits. The authors are applying these findings to laser-induced (LISW) brain injury. The authors need to establish the relevance of LISW to blast overpressure.

In general, the introduction is too short. I suggest the following:

1) Include literature that compared the common data elements of LISW and blast shock wave.

2) Include literature that shows LISW can have the same effects as IEDs, microexplosions and blast/shock tubes to be able to make a strong case for calling this model of injury a blast model.

3) Cite work from other authors who have shown blast related oxidative stress and inflammation. A single sentence on BBB disruption, inflammation, and edema doesn’t do justice to the body of literature that exists on blast-related neurological deficits. Since the focus is on oxidative stress, the authors can include few sentences on ROS, RNS, free radicals etc. There should be referece to some prior work that shows behavioral/motor deficits after blast. Also ref # 7 focuses on the efficacy of increasing the cerebral volume in being protective against blast-induced TBI.

4) How have antioxidants helped against blast-TBI? Authors have mentioned free radical scavenging. What happens to the endogenous antioxidatve mechanisms after blast exposure(s)?

5) Why AA2G in particular? What preliminary studies or literature supports the use of this particular antioxidant?

6) Why rotarod test in particular?

“AA2G (Hayashibara Co., Okayama, Japan) was dissolved in distilled water and was administered daily per os (p.o.) to rats at 250 mg/kg/15mL/day (ascorbic acid: 125 mg/kg/day) after LISW application for a total of 7 days (LISW + post AA2G group).” Did the animals receive ascorbic acid and its derivative (AA2G)? Please clarify the dosage and the daily frequency (once/day?) of what was administered.

Animal grouping: The description is a bit confusing. It has been described differently in different parts of the manuscript. Figure 1 is helpful I understanding the grouping. It seems like a total of four animal groups were used, which are: 1) Control, 2) LISW, 3) LISW+pre AA2G, and 4) LISW+pre-and post- AA2G. Figure 1B lists LISW again. Did the experimental design have a total of 20 animals in LISW group (combining Figures 1 A and B)? If so, why?

Why was rotarod testing done at 1 day prior to injury in some groups and 3 days prior to injury in other groups?

Generation and Characteristics of LISW: The shock wave generated by lasers (Fig 2) lack the underpressure that’s seen in the primary blast wave. It seems the authors are establishing LISW as a model of simulated blast injury. The authors need to provide more details (common data elements) of the pressure profile itself. Figure 2B- What are the characteristics after the 2 microsecond period? Are there any reflections or is it a flat line? Is the aim here to simulate primary blast wave? One has to make a really strong case to compare LISWs to blast explosions- otherwise the title is misleading. The application of the injury is very focal which is in contrast to blast injury (that’s diffused). Simulated blast pressures include static and dynamic pressures, which is lacking in the LISW model of injury. I strongly suggest changing the title of the manuscript to indicate this distinction clear.

“Although peak pressures and durations of LISWs are roughly three orders of magnitude higher and three orders of magnitude shorter than those of medically relevant actual explosions, respectively (28), impulse (time-integrated positive pressures) of actual explosions and LISWs used in this study are in the same order.” Please clarify. What blast pressures do they intend to compare the generated LISWs. If the comparison is made with mild blast intensities (10-15 PSI), then an increase by three orders of magnitude in peak intensity would make sense. What are the impulse values of the positive phase of LISW pressure profile?

“… impulse (time-integrated positive pressures) of actual explosions and LISWs used in this study are in the same order: The comment on impulse may not be correct. Please provide relevant impulse values from the literature.

“The laser spot size and fluence on the target were kept constant at 3 mm and 2.4 J/cm2, respectively, producing an LISW with an impulse of 54 Pa•s.” Please provide more details of the peak, duration, frequency, power, etc?

In general, LISW model as presented here may not be comparable to blast explosions. One would err on the side of caution and call this model simply as LISW model instead of clubbing it under blast. There isn’t enough evidence presented in the manuscript to support the author’s claim.

Line 155: rpm/min- correct to rpm

Statistics: Was the data checked for normal distribution?

Results:

Figure 3: Why are the sample sizes small? N= 4? With such small sample size, what’s of the power of the design? Is there any explanation for TNF-α being insignificantly different between control and injury groups?

Figures 4A and 4B should include the control group as well.

It’s been shown that post-LISW administration of AA2G is not beneficial in circumventing the oxidative stress induced by the injury. Pre+post- AA2G administration seems to be beneficial. Is it possible that simply the preemptive treatment with AA2G might be enough to inhibit the progression of oxidative stress? Why haven’t authors chosen to study only the prophylactic properties of the antioxidant? Please provide a justification. Combining the pre-and post- application in the same group of animals doesn’t necessarily rule out the therapeutic properties of AA2G- it suggests a combined prophylactic and therapeutic potential of AA2G.

6. PLOS authors have the option to publish the peer review history of their article (what does this mean?). If published, this will include your full peer review and any attached files.

Reviewer #1: Yes: Pinar Kuru Bektasoglu

Reviewer #2: Yes: Usmah Kawoos

---

## [Author Response · Author response to Decision Letter 0]

30 Dec 2019

Response to Reviewer #2:

 We would like to express our sincerest gratitude to the reviewer who identified areas of the manuscript that required revisions or modifications, and also for their insightful comments that have helped us to significantly improve our manuscript. We believe that laser-induced shock waves (LISWs) could be a useful tool to investigate the primary mechanisms of blast-related traumatic brain injury (bTBI) because they are spatially defined and are not accompanied by dynamic pressure, enabling site-selective analysis of primary injury mechanism. However, as highlighted by the reviewer, there are considerable differences in the characteristics of LISWs and actual blasts. Thus, the present animal model was changed and renamed as “LISW-induced TBI model,” rather than a “blast TBI model”. We have clearly stated our rationale for using this model (impulse mimicking) in the revised manuscript and have also discussed the LISW conditions compared with blasts as previously described by Mishra et al. (Sci. Rep., 2016). Our specific responses to each comment are detailed below. 

Comment 1. “Lines 62-65: Notably, most of the bTBI patients lack any external physical evidences and abnormalities detected by conventional imaging devices, but in the chronic phase, they develop persistent physiological and psychological changes associated with higher-order brain dysfunction, which is called blast-related mild TBI (bmTBI) (4, 5).”

In reference to this statement, authors should specify that most of the bTBI patients who are exposed to low or mild blast exposure may not show signs of overt pathology in the acute phase. Mild blast should not be confused with mild TBI.

Response: Thank you for your comment. We have revised the following text (page 5, lines 62–66) from “Notably, most of the bTBI patients lack any external physical evidences and abnormalities detected by conventional imaging devices, but in the chronic phase, they develop persistent physiological and psychological changes associated with higher-order brain dysfunction, which is called blast-related mild TBI (bmTBI) [4, 5]” to “Notably, most bTBI patients who are exposed to low or mild blast exposure lack any external physical evidences and abnormalities detected by conventional imaging devices; therefore, they are being diagnosed as having blast-related mild TBI (bmTBI). However, during the chronic phase, they develop persistent physiological and psychological changes associated with higher-order brain dysfunction during the chronic phase [4, 5].”

Comment 2. Lines 66-70: The cited work primarily evaluated the effects of shock wave (blast overpressure and/or underpressure) on neuropathology, oxidative stress, and the applicability of antioxidants in ameliorating the behavioral deficits. The authors are applying these findings to laser-induced (LISW) brain injury. The authors need to establish the relevance of LISW to blast overpressure.

Response: As described above, we have revised the manuscript considering the difference in characteristics between LISWs and actual blasts. In the present study, we found that topical application of LISWs to the rat brain caused oxidative stress and motor dysfunctions. There are some similarities in symptoms with those reported for mild bTBI rodent models using a shock tube [37, 38].

2-1) Include literature that compared the common data elements of LISW and blast shock wave.

Response: We have included new references [37] and [38] to discuss the LISW conditions used in this study compared with blast shock waves.

2-2) Include literature that shows LISW can have the same effects as IEDs, microexplosions and blast/shock tubes to be able to make a strong case for calling this model of injury a blast model.

Response: We have renamed the present model to an “LISW-induced TBI model,” rather than a “blast TBI model” to account for the characteristic differences between LISWs and actual blasts. In the present study, we assumed that the impulse was the primary parameter for determining brain injury. A description of the assumption has been included to the Materials and Methods section. While the impulse of LISWs used in the present study was low (54 Pa•s), we were unable to demonstrate that the TBI in our model was mild. Thus, the word “mild” has been deleted. 

2-3) Cite work from other authors who have shown blast related oxidative stress and inflammation. A single sentence on BBB disruption, inflammation, and edema doesn’t do justice to the body of literature that exists on blast-related neurological deficits. Since the focus is on oxidative stress, the authors can include few sentences on ROS, RNS, free radicals etc. There should be referece to some prior work that shows behavioral/motor deficits after blast. Also ref # 7 focuses on the efficacy of increasing the cerebral volume in being protective against blast-induced TBI.

Response: We apologize for the lack of a detailed explanation about blast-related oxidative stress. As per the reviewer’s comment, we have included the following text (page 5, line 16 to page 6, line 10): “Oxidative stress has been implicated in multiple models of TBI [12, 13] and is mainly induced by reactive oxidative species (ROS) such as superoxide, hydroxyl radical, and hydrogen peroxide [14, 15]. While basal levels of ROS are present during normal redox reactions and the electron transport chain, excess amounts, as seen after injury, can be harmful. NADPH oxidase (NOX) is a multi-subunit enzyme that catalyzes the formation of superoxide radicals from available molecular oxygen. NOX is upregulated in multiple brain regions following blast injury, and neurons maximally contributes to a higher increase in the hippocampus compared with other neural cells [16, 17]. Moreover, oxidative stress also contributes to enhanced BBB permeability during bTBI via a pathway that involves increased matrix metalloproteinase activation [18].”

2-4) How have antioxidants helped against blast-TBI? Authors have mentioned free radical scavenging. What happens to the endogenous antioxidatve mechanisms after blast exposure(s)?

Response: As the reviewer has pointed out, the mechanisms of endogenous antioxidation are very important. Therefore, we have added the following text (page 6, lines 11–16 and page 7, lines 1-3): “Hydrogen sulfide, which is one of the antioxidants, also affected the controlled cortical impact injury in rats, leading to improved neurologic dysfunction, increased activities of endogenous antioxidant enzymes (superoxide dismutase and catalase), decreased levels of oxidative products (malondialdehyde and 8-iso-prostaglandin F2α), increased BBB permeability, and attenuation of brain edema. Furthermore, the use of the KATP channel blocker, 5-hydroxydecanoate, further demonstrated activation of mitochondrial adenosine triphosphate sensitive potassium channels, and oxidative stress was reduced following treatment with exogenous hydrogen sulfide [19, 20].” 

2-5) Why AA2G in particular? What preliminary studies or literature supports the use of this particular antioxidant?

Response: Thank you for highlighting this. We apologize for not acknowledging the efficacy of AA2G. We have included the following text (page 26, lines 6–10): “Further, AA2G is an ascorbic acid derivative that is stable in aqueous solution and barely induces cellular toxicity in cultured stem cells, unlike ascorbic acid [56]. In addition, AA2G has protective effects against Helicobacter pylori infection in gastric epithelial cells [30] and confers protective effects on human sperm motility preservation through the freeze–thaw cycle [57] .”

2-6) Why rotarod test in particular?

Response: We apologize for not explaining our rationale for selecting the rotarod test. As per the reviewer’s comment, we have added the following text (page 23 lines 5–12): “Studlack et al. previously used beam walk, accelerating rotarod, rearing, open field, elevated plus maze, and light–dark box task tests in bTBI injury model rats to evaluate their motor function and anxiety-related behavior. They observed that the accelerating rotarod test exhibited significant differences between the bTBI and control groups from day 1 to day 14 post injury [45]. We considered that the prolonged symptoms were suitable for evaluating the efficacy of AA2G and confirmed that the accelerating rotarod test exhibited significant differences between the LISW and control groups in the present study.” 

Comment 3. “AA2G (Hayashibara Co., Okayama, Japan) was dissolved in distilled water and was administered daily per os (p.o.) to rats at 250 mg/kg/15mL/day (ascorbic acid: 125 mg/kg/day) after LISW application for a total of 7 days (LISW + post AA2G group).” Did the animals receive ascorbic acid and its derivative (AA2G)? Please clarify the dosage and the daily frequency (once/day?) of what was administered.

Response: We apologize for this confusion. In the present study, 250 mg of AA2G contained 125 mg of ascorbic acid. AA2G was administered daily in the drinking water. We modulated the concentration of AA2G in the drinking water by measuring the daily water intake to administer a dose of 250 mg/kg/day of AA2G to the rats. 

 We have revised the following text in lines 12–15 on page 8 from “AA2G (Hayashibara Co., Okayama, Japan) was dissolved in distilled water and was administered daily per os (p.o.) to rats at 250 mg/kg/15mL/day (ascorbic acid: 125 mg/kg/day) after LISW application for a total of 7 days (LISW + post AA2G group)” to “AA2G (Hayashibara Co., Okayama, Japan) was dissolved in distilled drinking water and was administered daily per os (p.o.) to rats at a dose of 250 mg/kg/day (250 mg of AA2G contained 125 mg/kg/day of ascorbic acid) after LISW application for a total of 7 days (LISW + post AA2G group).”

Comment 4. Animal grouping: The description is a bit confusing. It has been described differently in different parts of the manuscript. Figure 1 is helpful I understanding the grouping. It seems like a total of four animal groups were used, which are: 1) Control, 2) LISW, 3) LISW+pre AA2G, and 4) LISW+pre-and post- AA2G. Figure 1B lists LISW again. Did the experimental design have a total of 20 animals in LISW group (combining Figures 1 A and B)? If so, why?

Response: Thank you for highlighting this. The experimental design included a total of 20 animals in the LISW group. We observed no significant difference between the LISW and LISW + post-AA2G groups, and we could only add pre- and post-AA2G groups as further experiment. We would like to compare LISW group and pre- and post-AA2G group under the same conditions because ascorbic acid levels in the tissue were quite unstable. Therefore, we could perform additional experiments (assessment of motor function, oxidative stress, ascorbic acid levels, and DHE staining) to compare the control, LISW, and LISW + pre- and post-groups. 

 

Comment 5. Why was rotarod testing done at 1 day prior to injury in some groups and 3 days prior to injury in other groups?

Response: We apologize for the confusion in the description of the method. We have revised Fig. 1 and the following text in line 11 on page 9 from “their motor function was assessed using a rotarod apparatus 1 day before and 7 days after the procedures” to “their motor function was assessed using a rotarod apparatus 3 day before and 7 days after the procedures.”

Figure 1 (revised)

Comment 6. Generation and Characteristics of LISW: The shock wave generated by lasers (Fig 2) lack the underpressure that’s seen in the primary blast wave. It seems the authors are establishing LISW as a model of simulated blast injury. The authors need to provide more details (common data elements) of the pressure profile itself. Figure 2B- What are the characteristics after the 2 microsecond period? Are there any reflections or is it a flat line? Is the aim here to simulate primary blast wave? One has to make a really strong case to compare LISWs to blast explosions- otherwise the title is misleading. The application of the injury is very focal which is in contrast to blast injury (that’s diffused). Simulated blast pressures include static and dynamic pressures, which is lacking in the LISW model of injury. I strongly suggest changing the title of the manuscript to indicate this distinction clear.

Response: Typical LISWs are dominated by positive pressure, although negative pressure can be generated under certain conditions. In the present study, generated LISWs did not interact with the external environment and were directly propagated into the rat brain. There were no pressure signals beyond the time range shown in Fig. 2B (>2 μs). LISWs are spatially confined and lack dynamic pressure component resulting in accelerating injury (the tertiary mechanism). We believe that these characteristics of LISWs provide a unique tool that can be used to analyze the primary mechanisms of bTBI. However, we agree that LISWs and accrual blasts differ in characteristics. Thus, we renamed our model as an “LISW-induced TBI model”, rather than a “blast TBI model”. The LISW impulse used was as low as 54 Pa•s, but the peak overpressure was high. Therefore, we have changed the title to remove the word “mild”. 

Comment 7. “Although peak pressures and durations of LISWs are roughly three orders of magnitude higher and three orders of magnitude shorter than those of medically relevant actual explosions, respectively (28), impulse (time-integrated positive pressures) of actual explosions and LISWs used in this study are in the same order.” Please clarify. What blast pressures do they intend to compare the generated LISWs. If the comparison is made with mild blast intensities (10-15 PSI), then an increase by three orders of magnitude in peak intensity would make sense. What are the impulse values of the positive phase of LISW pressure profile?

Comment 8. “… impulse (time-integrated positive pressures) of actual explosions and LISWs used in this study are in the same order: The comment on impulse may not be correct. Please provide relevant impulse values from the literature.

Response:

Responses to Comments 7 & 8:

 We have included two references (new ref. [37], [38]) to describe the pressure characteristics of improvised explosive devices (IEDs), and we have described the LISW conditions with reference to that reported by Mishra et al. [17] in the Materials and Methods section. The phrases “three orders of magnitude” has been corrected and changed to “two to three orders of magnitude.” We assumed that the impulse was the primary parameter to determine brain injury and used LISWs replicating the impulse of a typical IED.

 In accordance with the reviewer’s comment, we have included the following text (page 10, line 3 to page 11, line 5): “No dynamic pressure is produced in the generation of LISWs; therefore, the effects of acceleration or displacement can be excluded in animal studies, thus enabling analysis of the primary mechanism of bTBI. Positive pressure duration (hereafter simply termed duration) is an important parameter to examine shock wave to brain interactions. The duration of a typical IED explosion-related shock wave ranges from 200 μs to several milliseconds [37, 38]; however, the interaction of the human brain with such IED-related shock waves, especially brain to skull boundary effects, cannot be reproduced in the brains of small animals when using similar duration shock waves. Therefore, a scaling law that considers anatomical differences between human and animal brains should be used. However, few studies have focused on this important issue. In the present study, we assumed that the impulse (time-integrated positive pressure component) of the shock wave was the primary parameter for determining brain injury. Alley previously estimated impulses of various types of IEDs as a function of propagation distance ranging from ~6.9 to ~100 Pa•s for propagation distances of 1–10 m [37]. Although the duration of LISWs is approximately two to three orders of magnitude shorter than that of typical IED shock waves, their impulses can be easily controlled by changing the laser fluence and hence the peak pressure of LISW. IED impulses within this range can be replicated using LISWs. The limitations associated with this assumption are described in Discussion and Limitations section.” 

 We have also added the following text (page 12, lines 6-12): “Mishra et al. examined physiological and pathological changes, as well as changes in blood-borne biomarkers in rats exposed to blasts with a wide range of overpressure and impulses using a shock tube [17]. They determined the conditions under which mild, moderate, severe, and lethal bTBIs were induced, and postulated that mild bTBI was caused by blasts with ≤145 kPa, with a corresponding impulse of 250 Pa•s. The impulse used in this study (54 Pa•s) was within this range.” 

Comment 9. “The laser spot size and fluence on the target were kept constant at 3 mm and 2.4 J/cm2, respectively, producing an LISW with an impulse of 54 Pa•s.” Please provide more details of the peak, duration, frequency, power, etc?

Response: The temporal pressure profile of the used LISW is shown in Fig. 2B (blue line). A single pulse of LISW was applied to each site on the skull. 

Comment 10. In general, LISW model as presented here may not be comparable to blast explosions. One would err on the side of caution and call this model simply as LISW model instead of clubbing it under blast. There isn’t enough evidence presented in the manuscript to support the author’s claim.

Response: We agree with the reviewer’s comment and have changed the name of our model to an “LISW-induced TBI model”.

Comment 11. Line 155: rpm/min- correct to rpm

Response: We apologize for this error. We have changed “rpm/min” to “rpm” in line 5 on page 14.

Comment 12. Statistics: Was the data checked for normal distribution?

Response: We apologize for not verifying the normal distribution of our data. We performed Kolmogorov–Smirnov test to evaluate whether our data was normally distributed and confirmed that the data in Figs. 4 and 5 were normally distributed. In contrast, we were unable to perform a Kolmogorov–Smirnov test on the data in Fig. 3 because of the small sample size. Therefore, we performed Mann–Whitney test instead of a Student’s t test to analyze the data in Fig. 3. We have revised the following text in line 9 on page 17 from “Differences between two experimental groups were assessed using Student’s t-test” to “Differences between two experimental groups were assessed using Mann–Whitney test.” 

Comment 13. Figure 3: Why are the sample sizes small? N= 4? With such small sample size, what’s of the power of the design? Is there any explanation for TNF-α being insignificantly different between control and injury groups?

Response: As highlighted by the reviewer, the small sample sizes in Fig. 3 meant that we were unable to draw a significant conclusion. However, this experiment was a preliminary step in our study and was not designed to be conclusive. We did not focus on inflammatory cytokines in this figure, but instead, we examined oxidative stress. This was performed to demonstrate that oxidative stress in the brain may increase and TNF-α may not significantly increase. Furthermore, there was a significant difference in the oxidative stress levels between the control and LISW groups in this setting, and there was no need to add additional experiments. We have added the following text (page 18, line 10): “at least in the limited cases.” 

 

Comment 14. Figures 4A and 4B should include the control group as well.

Response: Thank you for highlighting this error. We have included the control group to Figs. 4A and 4B. Statistical analyses were conducted using one-way ANOVA with Tukey HSD post hoc test. We found that there was no significant difference between the LISW and post AA2G groups. 

Fig 4 (revised)

Comment 15. It’s been shown that post-LISW administration of AA2G is not beneficial in circumventing the oxidative stress induced by the injury. Pre+post- AA2G administration seems to be beneficial. Is it possible that simply the preemptive treatment with AA2G might be enough to inhibit the progression of oxidative stress? Why haven’t authors chosen to study only the prophylactic properties of the antioxidant? Please provide a justification. Combining the pre-and post- application in the same group of animals doesn’t necessarily rule out the therapeutic properties of AA2G- it suggests a combined prophylactic and therapeutic potential of AA2G.

Response: We apologize for not including a detailed description of the AA2G administration. In our previous study on the effects of ascorbic acid administration in a mouse model of radiation-induced gastrointestinal damage (Ito et al, Int J Mol Sci 2013, 14 19168-19635), pre-treatment alone or post-treatment alone did not improve the survival of the mice. Nevertheless, in the present study, pre and post-treatment significantly improved the survival of the mice. It is possible that scavenging of ROS generated immediately after radiation by pretreatment with ascorbic acid may be necessary to improve the survival of irradiated mice. However, additional post-treatment with ascorbic acid also may be indispensable for further improving the survival due to late or ongoing damage by oxidative stress. Oxidative stress may also be an important factor in bmTBI injury; therefore, pre + post-treatment by AA2G may have been more effective than pre-treatment alone. For this reason, we chose pre + post-treatment model instead of pre-treatment model. 

We have included the following text (page 27, lines 10-16): “In these studies, pre-treatment or post-treatment alone did not improve the survival of the mouse model; however, pre- and post-treatment significantly improved the survival. It is possible that scavenging of ROS generated immediately after radiation by boosting the pretreatment with ascorbic acid may be necessary to improve the survival of irradiated mice. However, additional post-treatment with ascorbic acid may also be indispensable to further improve the survival due to late or ongoing damage by oxidative stress.”

We hope that our responses and revisions have appropriately addressed your comments, and that our revised manuscript is now suitable for publication. We look forward to hearing from you. Please let us know if you have any further questions or requirements to improve our manuscript.

---

## [Decision Letter · Decision Letter 1]

10 Mar 2020

Oral ascorbic acid 2-glucoside prevents coordination disorder induced via laser-induced shock waves in rat brain

PONE-D-19-25644R1

Dear Dr. Maekawa,

We are pleased to inform you that your manuscript has been judged scientifically suitable for publication and will be formally accepted for publication once it complies with all outstanding technical requirements.

With kind regards,

Alfred S Lewin, Ph.D.

Section Editor

PLOS ONE

Additional Editor Comments (optional):

Reviewers' comments:

Reviewer's Responses to Questions

**Comments to the Author**

1. If the authors have adequately addressed your comments raised in a previous round of review and you feel that this manuscript is now acceptable for publication, you may indicate that here to bypass the “Comments to the Author” section, enter your conflict of interest statement in the “Confidential to Editor” section, and submit your "Accept" recommendation.

Reviewer #2: All comments have been addressed

2. Is the manuscript technically sound, and do the data support the conclusions?

Reviewer #2: Yes

3. Has the statistical analysis been performed appropriately and rigorously? 

Reviewer #2: Yes

4. Have the authors made all data underlying the findings in their manuscript fully available?

Reviewer #2: Yes

5. Is the manuscript presented in an intelligible fashion and written in standard English?

Reviewer #2: Yes

6. Review Comments to the Author

Reviewer #2: The authors gave satisfactorily revised the manuscript. Needs minor editorial corrections, but otherwise all comments have been addressed.

7. PLOS authors have the option to publish the peer review history of their article (what does this mean?). If published, this will include your full peer review and any attached files.

Reviewer #2: Yes: Usmah Kawoos

---

## [Editor Report · Acceptance letter]

16 Mar 2020

PONE-D-19-25644R1 

Oral ascorbic acid 2-glucoside prevents coordination disorder induced via laser-induced shock waves in rat brain 

Dear Dr. Maekawa:

I am pleased to inform you that your manuscript has been deemed suitable for publication in PLOS ONE. Congratulations! Your manuscript is now with our production department. 

With kind regards,

on behalf of

Dr. Alfred S Lewin 

Section Editor

PLOS ONE